# Improved Antithrombotic Activity and Diminished Bleeding Side Effect of a PEGylated α_IIb_β_3_ Antagonist, Disintegrin

**DOI:** 10.3390/toxins12070426

**Published:** 2020-06-28

**Authors:** Yu-Ju Kuo, Yao Tsung Chang, Ching-Hu Chung, Woei-Jer Chuang, Tur-Fu Huang

**Affiliations:** 1Department of Medicine, Mackay Medical College, New Taipei City 25245, Taiwan; d01443001@ntu.edu.tw (Y.-J.K.); chchung@mmc.edu.tw (C.-H.C.); 2Department of Biochemistry, National Cheng Kung University Medical College, Tainan 70101, Taiwan; ytchang.ncku@gmail.com; 3Graduate Institute of Pharmacology, College of Medicine, National Taiwan University, Taipei 10051, Taiwan

**Keywords:** PEGylation, polymer conjugation, disintegrins, stability, protein therapeutics, pharmacokinetic, antiplatelet agent, antithrombotics, safety profiles

## Abstract

Polymer polyethylene glycol (PEG), or PEGylation of polypeptides improves protein drug stability by decreasing degradation and reducing renal clearance. To produce a pharmaceutical disintegrin derivative, the N-terminal PEGylation technique was used to modify the disintegrin derivative [KGDRR]trimucrin for favorable safety, pharmacokinetic profiles, and antithrombotic efficacy. We compared intact [KGDRR]trimucrin (RR) and PEGylated KGDRR (PEG-RR) by in vitro and in vivo systems for their antithrombotic activities. The activity of platelet aggregation inhibition and the bleeding tendency side effect were also investigated. PEG-RR exhibited optimal potency in inhibiting platelet aggregation of human/mouse platelet-rich plasma activated by collagen or ADP with a lower IC_50_ than the intact derivative RR. In the illumination-induced mesenteric venous thrombosis model, RR and PEG-RR efficaciously prevented occlusive thrombosis in a dose-dependent manner. In rotational thromboelastometry assay, PEG-RR did not induce hypocoagulation in human whole blood even given at a higher concentration (30 μg/mL), while RR slightly prolonged clotting time. However, RR and PEG-RR were not associated with severe thrombocytopenia or bleeding in FcγRIIa-transgenic mice at equally efficacious antithrombotic dosages. We also found the in vivo half-life of PEGylation was longer than RR (RR: 15.65 h vs. PEG-RR: 20.45 h). In conclusion, injectable PEG-RR with prolonged half-life and decreased bleeding risk is a safer anti-thrombotic agent for long-acting treatment of thrombus diseases.

## 1. Introduction

Integrins, a family of cell adhesion receptors, are essential for cell migration and invasion, mediating both cell/cell and cell/matrix interactions [1]. Integrins exist as an α:β heterodimeric complex of transmembrane proteins [2]. The most abundant integrin in platelets is integrin α_IIb_β_3_. In addition, integrin α_IIb_β_3_ is a receptor for adhesive proteins, including fibrinogen, fibronectin, and vitronectin. By binding to the ligand, integrin mediates platelet adhere to injured vessel walls and platelet aggregation, which is important for the maintenance of hemostasis, preventing excessive bleeding [3].

Disintegrins are α_IIb_β_3_ antagonist and potential antithrombotic agents of platelet aggregation found in snake venom [4]. Disintegrins were originally known as low molecular weight, cysteine-rich venom proteins that contain an RGD (Arg-Gly-Asp) or KGD (Lys-Gly-Asp) loop maintained by specific disulfide bridges [5]. RGD mimetic agents, such as eptifibatide and tirofiban, are clinically used to prevent thrombosis in patients undergoing percutaneous coronary angioplasty and stenting. However, increased bleeding risk limits RGD mimetic drugs from being used in patients undergoing percutaneous coronary intervention [6]. Furthermore, their short half-life limits therapeutic efficacy and requires a frequent administration regimen to against rapid clearance from the circulation [7], thus require the use of high doses to maintain therapeutic efficacy. In our previous studies, we discovered a disintegrin purified from *Trimeresurus mucrosquamatus* snake venom, trimucrin (i.e., TMV-7), consists of 73 amino acid residues including an RGD sequence at position 51–53 [8,9]. The RGD-containing disintegrin trimucrin, like the RGD mimetics eptifibatide, inhibits agonist-induced platelet aggregation and thrombus formation through blockade of integrin α_IIb_β_3_ [8]. In this study, we found that the potent antithrombotic α_IIb_β_3_ antagonist trimucrin has lower bleeding risk and longer circulating half-life (t1/2) than eptifibatide, however, as being used at higher doses, trimucrin still slightly prolonged tail-bleeding time. Therefore, a long acting antithrombotic agent without causing a bleeding side effect is under active investigation.

PEGylation is one of the successful strategies to overcome these disadvantages by conjugating polyethylene glycol (PEG) to drugs [10]. The water-soluble biocompatible PEG (ethylene glycol (-CH2-CH2-O-)) was used to conjugate surfaces of biomedical devices or drugs. Based on its adsorption-resistance mechanism, the PEG conjugation significantly enhances their hydrodynamic size via its hydration effect and prevents rapid renal clearance to prolong the half-life [11]. Hydrated PEG was also reported to mitigate immunogenicity, antigenicity, and toxicity through shielding antigenic epitopes from immune system recognition [12,13]. Currently, PEG is one of a limited number of synthetic polymers generally regarded as safe by the United States Food and Drug Administration (US FDA) for internal administration, and more than 20 PEGylated drugs are currently in clinical trials [14].

To address these bleeding and low stability issues, we mutated the RGD-domain of trimucrin from ^50^ARGDNP^55^ to ^50^AKGDRR^55^, and found that the safety index of the disintegrin derivative [KGDRR]trimucrin (RR) is raised to 70-times higher than trimucrin. In addition, we constructed a new RR derivative using an established N-terminal PEGylation technique. The conjugation of RR with PEG was considered to improve its pharmacological activities and pharmaceutical advantages. In the present study, we compared pharmacokinetic characterization between native RR and PEGylated RR (PEG-RR) and investigated their in vivo anti-thrombosis efficacy in vessel injury-induced thrombosis model. In addition, we used monoclonal AP2 as a platform [8,15] to evaluate the adverse reaction of bleeding upon administration of RR and PEG-RR in vitro, and further investigated their tendency to cause thrombocytopenia in FcγRIIA-transgenic mouse model ex vivo. In conclusion, this study provides a picture of how to achieve a stable formulation with minimal bleeding side effects and long-circulating properties.

## 2. Results

### 2.1. Modification of Trimucrin Derivative RR with PEGylation

We previously reported that safer antithrombotic α_IIb_β_3_ antagonists do not increase bleeding risk in vivo at efficacious antithrombotic doses [8]; however, as intravenously administrated trimucrin into mice with 20-times higher dosage (5 mg/kg), trimucrin significantly prolonged tail bleeding times from 76.50 to 190.20 s (*p* < 0.05; Table 1). We mutated the RGD loop of trimucrin sequence [8] from ^50^ARGDAR^55^ to ^50^AKGDRR^55^ to improve a better safety profile (Table 1) and performed an arginine-to-lysine substitution (R51K) to enhance its specificity for α_IIb_β_3_. The trimucrin derivative was named RR and exhibited higher potency in inhibiting collagen-induced platelet aggregation in human platelet suspension, and its safety index is 935-fold and 144-fold higher than eptifibatide and trimucrin, respectively (Table 1). The trimucrin derivative RR was further conjugated with polyethylene glycol (PEG).

The purity and molecular weight of RR and PEG-RR were checked by 12% sodium dodecyl sulfate-polyacrylamide gel electrophoresis (SDS-PAGE) and mass spectrometry. As shown in Figure 1A, RR (15 μg) and PEG-RR (3 μg) were subjected to SDS-PAGE and stained with Coomassie brilliant blue, we found that the mobility of RR and PEG-RR was similar under non-reduced and reduced conditions, and their molecular weight was approximately 8 and 40 kDa, respectively. The exact molecular weight of purified RR and the average molecular weight of PEG-RR were confirmed by mass spectrometry (Figure 1B,C) as being 8035.48 Da and 40340.09 Da, respectively. The molecular mass of RR determined by mass spectrometry was in excellent agreement with the calculated value of 8036.07 Da (0.59 Da deviation).

### 2.2. Inhibitory Efficacy of RR and PEG-RR on Platelet Aggregation

PEG-RR exhibited more potent anti-platelet activity than intact RR and eptifibatide in both human and mouse platelets. In human platelet suspension (PS), like eptifibatide, both the KGD-containing disintegrin RR and PEG-RR caused concentration-dependent inhibition on the platelet aggregation induced by thrombin (0.1 U/mL, Figure 2A) or collagen (10 μg/mL, Table 1). The IC_50_ values of PEG-RR were approximately 11.85 nM and 8.6 nM, respectively (*n* = 6); the IC_50_ values of RR were approximately 28.84 nM and 36.9 nM, respectively (*n* = 6); and the IC_50_ values of eptifibatide were approximately 179.09 nM and 625.0 nM, respectively (*n* = 6). Furthermore, RR and PEG-RR also blocked the platelet aggregation caused by ADP (20 μM) in a concentration-dependent manner (Figure 2B,C). The IC_50_ values of PEG-RR were 11.32 nM and 5.27 nM, respectively (*n* = 7); the IC_50_ values of RR were approximately 46.39 nM and 31.34 nM, respectively (*n* = 7); and the IC_50_ values of eptifibatide were approximately 492.81 nM and 877.45 nM, respectively (*n* = 7). Overall, PEG-RR was 3–4 times more potent than RR and 50–150 times more potent than eptifibatide in inhibiting agonist-induced platelet aggregation.

### 2.3. Comparison of the ex vivo Antiplatelet Activity between RR and PEG-RR

To evaluate whether RR and PEG-RR have comparable effect on mouse platelets as human platelets, we assessed the ex vivo antiplatelet action of RR and PEG-RR in mouse platelet-rich plasma (PRP). Mice were intravenously treated with agents, and then blood samples were collected by intracardiac puncture. We centrifuged the whole mouse blood sample at 200× *g* for 4 min to obtain PRP, and then added inducer ADP (left, 20 μM) or collagen (right, 10 μg/mL) to trigger platelet aggregation. Platelet aggregation was measured by the turbidimetric method (ΔT) using a platelet aggregometer. As shown in Figure 3A,B, either RR or PEG-RR efficaciously inhibited platelet aggregation induced by 20 μM ADP or 10 μg/mL collagen of mouse PRP. PEG-RR exhibited higher ex vivo potency as compared to RR and eptifibatide. Furthermore, the collagen-induced initial platelet shape change in PRP was not affected by both RR and PEG-RR. Taken together, these results indicated that the chemical conjugation not only conserves the full activity of native RR but also enhances its efficacy on platelet aggregation.

### 2.4. Effects of RR and PEG-RR on In Vivo Mouse Bleeding Tendency and Human Physiological Hemostatic Function in ROTEM Assays

Since the condition in which exposure to current RGD-mimetic drugs (i.e., eptifibatide and tirofiban) leads to the destruction of circulating platelets, often accompanied by bleeding symptoms [6], it is necessary to consider whether administration of intact RR and PEG-RR cause such an adverse reaction. It has been well established that an Fc receptor for IgG, FcγRIIA, is expressed on human platelets/macrophages and plays a vital role in immune-mediated thrombocytopenia [16,17]. In this study, we compared RR and PEG-RR with eptifibatide on platelet counts and tail-bleeding times in the FcγRIIa-transgenic mouse model [16]. As shown in Figure 3C,D, at efficacious antiplatelet doses, eptifibatide significantly decreased platelet counts after 5 and 10 h in transgenic mice (Figure 3C).

Consistent with this result, the bleeding times of eptifibatide-treated FcγRIIa-transgenic mice were time-dependently prolonged with decreases in platelet counts (Figure 3D). In contrast, neither RR nor PEG-RR caused thrombocytopenia or prolonged bleeding times in FcγRIIa-transgenic mice at efficacious antiplatelet doses. Since bleeding risks limit the use and doses of current integrin α_IIb_β_3_ antagonists, we further examined the safety margin of PEG-RR. As shown in Figure 3D, we increased the administrated dosage of PEG-RR to 10-fold higher (from 0.1 to 1 mg/kg) and found that PEG-RR, unlike current antithrombotics, did not prolong tail bleeding times even at an extremely high dose, indicating that PEG-RR exhibit a favorable safety profile.

We further evaluated the effect of RR and PEG-RR on primary hemostatic function in human whole blood by rotational-thromboelastometry (ROTEM), which is a point-of-care method of assessing the efficiency of blood coagulation, clot strength, and the risk factors for bleeding, and was mainly utilized within the perioperative scenario [18,19]. As shown in Table 1, the ROTEM test was intrinsically activated by ellagic acid, and the clotting time (CT) represented the initial coagulation rate; clot formation time (CFT) and α angle showed a good correlation to fibrin polymerization and aPTT (activated partial thromboplastin time)/PT (prothrombin time); maximum clot firmness (MCF) was correlated to platelet count; and above four variables were incorporated into a coagulation-index (CI; Figure 4; or overall assessment of coagulation). Tirofiban and eptifibatide concentration-dependently prolonged CT and CFT, and significantly decreased MCF, leading to a hypocoagulation state, while neither RR nor PEG-RR affected all ROTEM variables as compared to the control group. These results indicated that they have less influence on physiological hemostasis. Importantly, PEG-RR even exhibited similar safety at doses of 60-fold higher (30 μg/mL; Figure 4B) that required for efficacy, while the intact RR showed descending CI value.

### 2.5. Comparison of Anti-Thrombotic Activity between RR and PEG-RR in Illumination-Induced Thrombosis of Mouse Mesenteric Venules

We determined antithrombotic effects of RR and PEG-RR in vivo. The male Institute of Cancer Research (ICR) mice from Institute of Cancer Research were injected with increasing dosage of RR or PEG-RR 5 min before illumination-induced a venular wall injury, and then the occlusion was observed in illuminated mesenteric venules of mice and monitored by intravital microscopy. As shown in Figure 5, occlusion occurred in all saline treated mice around 30 sec after injury, and these injured vessels were fully blocked at around 180 sec and no recanalization was observed until the endpoint. By contrast, both RR and PEG-RR dose-dependently prevented occlusive thrombosis over 180 sec after vessel injury and even recovered blood flow after thrombus formation. These data provide direct evidence that RR and PEG-RR effectively prevent occlusive thrombosis.

### 2.6. In vivo Functional Half-Life (t1/2) of RR and PEG-RR in Inhibiting Platelet Aggregation

To evaluate the functional half-life of RR and PEG-RR, we examined the ex vivo antiplatelet activity at various time intervals (Figure 6). At equally efficacious dosage ex vivo, the calculated half-life (t1/2) of PEG-RR (0.1 mg/kg) in inhibiting collagen-induced platelet aggregation was 1.3 and 7.9 times longer than RR (0.125 mg/kg) and eptifibatide (0.18 mg/kg), respectively. Note that the inhibitory effect of PEG-RR (0.2 mg/kg) was sustained for longer than 64.2 h (t1/2 = 32.1 h), demonstrating that significantly prolonged serum presence of RR by PEGylation. This result indicated that eptifibatide might be rapidly cleared from the plasma after t1/2 of 2.55 h and revealed the potential of PEGylation to improve the pharmacokinetics of protein-based medicines.

## 3. Discussion

Several materials have been studied to extend the half-life and increase the activity of anti-thrombotic agents in the bloodstream, such as a polymer coating or drug carriers. Conjugation of drugs with PEG was the most common approach among these materials. PEG enhances drug solubility and limits renal filtration, proteolytic degradation, and immunogenicity [20]. The present study focused on development and evaluation of a long-acting anti-thrombosis agent through PEGylation of α_IIb_β_3_ antagonist. PEGylated rhodostomin was successfully modified with PEG in optimal condition as we reported previously [21]. In this study, a novel N-terminal PEGylation RR derivative was designed to improve the pharmaceutical entities. As the effective molecular weight of crude RR was around 8 kDa, the theoretical molecular weight of PEG-RR was around 38 kDa. The SDS-PAGE result was consistent with our expectation, suggesting that RR was successfully modified with PEG.

In our study, the in vitro antiplatelet activity of the PEGylated products was higher than that of purified RR in both human and mouse platelets. We further investigated their in vivo antithrombotic potency in the illumination-induced mesenteric venous thrombosis model. The efficacious dosage of RR and PEG-RR in prolonging the thrombus formation time was around 0.2 mg/kg, demonstrating the in vivo potent of PEG-RR is about 5-times higher based on molarity since the molecular weight of RR and PEG-RR are 8 kDa and 40 kDa, respectively. Furthermore, the reduction ranges of PEG-RR were much smaller than those of native RR. Around 24 h after the injection of native RR, its ex vivo antiplatelet activity was absent, while the antiplatelet activity of PEG-RR remained 50% inhibition of inducer-stimulated platelet aggregation. These results indicated that PEG-RR had long-circulating effects/prolonged biological half-life in vivo, and this effect may be associated with its clearance of glomerular filtration decreasing. PEG conjugation not only increased the size and molecular weight of RR but also improved its pharmacokinetics by protecting from enzymatic degradation, prolonging circulation life [22]. A PEG polymer linked to RR may create a bulky hydrophilic shield that could efficiently mask the conjugated RR from enzymatic digestion.

Since bleeding complication is the major clinical limitation of the current α_IIb_β_3_ antagonists in clinical use [6,23], we evaluated the bleeding tendency of RR and PEG-RR in the FcγRIIa-transgenic mouse model. Our present results indicated that both RR and PEG-RR neither decrease platelet counts nor prolong bleeding time at efficacious dosage in vivo, in contrast to current α_IIb_β_3_ antagonists. We further evaluated their bleeding tendency by assessing their viscoelastic properties of coagulation in whole blood via a ROTEM assay, which is commonly used in a clinic as a surrogate measurement of physiological hemostasis function [21]. Consistent with the observation in vivo, unlike the clinic used antithrombotics tirofiban and eptifibatide, RR and PEG-RR neither prolonged clotting time and clot formation time nor decreased maximum clot firmness (Table 1). Both RR and PEG-RR showed a favorable safety index at efficacious concentration. However, at doses of 60-fold higher (30 μg/mL; Figure 4B), PEG-RR even exhibited similar safety, while RR showed a slight decrease in the coagulation index, leading to a hypocoagulation state. In line with the above observations, PEG-RR exhibited more potent antiplatelet and antithrombotic effect, as well as a more stable pharmacokinetic profile and favorable safety profiles than native RR and current α_IIb_β_3_ antagonists.

Recently, new evidence has emerged on potentially important differences between thrombosis and hemostasis [24,25,26], thus raising the possibility of developing new antithrombotic drugs without increasing bleeding risk. Two potential strategies have been reported to achieve this end: (1) targeting factors that mediate the thrombus formation but do not affect the initial formation of the thrombus core [27] and (2) discovering inhibitors that bind to the extracellular domain of the integrin without causing receptor activation and conformational changes [28,29,30]. These strategies may hold the key to safer antithrombotic therapy.

## 4. Conclusions

The vascular thrombotic disease is a serious threat to human health that is the main cause of human disability and death. The present findings showed that PEG-modified RR exhibited a greater anti-thrombosis effect and pharmacokinetic characteristics than native RR and current antithrombotics. Furthermore, PEG-RR had prolonged half-life and favorable safety profile with no thrombocytopenia or bleeding side effects in vivo. These results suggest that the PEGylated RR is a novel anti-thrombosis agent for the long-acting treatment of thrombosis diseases. It might effectively improve the clinical symptoms of patients and reduce the overall morbidity and morbidity, acting as an ideal agent to treat vascular thrombotic disease in clinical settings. Taken together, these findings indicate that PEGylation enhances the pharmacological efficacy of α_IIb_β_3_ antagonists’ therapeutics due to increasing its half-life, and also are helpful for developing the second generation of α_IIb_β_3_ antagonist.

## 5. Materials and Methods

### 5.1. Materials

Prostaglandin E1 (PGE1; P5515), heparin (H3149), adenosine diphosphate (ADP; 01905), collagen (bovine tendon type I; 900722), human thrombin (605195), Coomassie blue R-250 (1.12553), SDS (71725), and BSA (A2153) were purchased from Sigma Chemical Co. (St. Louis, MO, USA). FITC-conjugated goat anti-mouse IgG (SC-2010) was obtained from Santa Cruz Biotechnology, Inc. (Santa Cruz, CA, USA).

### 5.2. Preparation of the PEG-RR

For the synthesis of the PEG-RR (Figure 1), the PEGylation was prepared as previously reported [24,25]. Firstly, RR mutant expressed in *P. pastoris* were purified to homogeneity by CaptoMMC chromatography and C_18_ reversed-phase HPLC. The purified RR protein was mixed with mPEG-aldehyde at a (protein: PEG) molar ratio of 1:8 in sodium acetate buffer (100 mM, pH 5.0) containing 20 mM sodium cyanoborohydride (NaCNBH3) at 25 °C, reacted for 2 h. Then the reaction mixtures of RR and PEG-RR were purified by MacroCap SP column using linear gradient from 0 to 1.0 M NaCl, 50 mM sodium acetate buffer at pH 4.0. The mono-PEGylated-RR was further purified to homogeneity by reversed phase C_18_ HPLC.

### 5.3. Mass Spectrometric Measurements

The molecular weight of RR protein was confirmed using the LTQ Orbitrap XL mass spectrometer equipped with a heated-electrospray ionization source (Thermo Fisher Scientific Inc., Waltham, MA, USA). The protein solution (100 μg/mL in 75% methanol with 30 mM formic acid and 10 mM triethylamine) was infused into the mass spectrometer by using a syringe pump at a flow rate of 3 μL/min to acquire full scan mass spectra. The electrospray voltage at the spraying needle was optimized at 3500 V. The molecular weight of protein was calculated by computer software Xcalibur that was provided by Thermo Fisher Scientific.

The molecular weight of PEG-RR was confirmed using MALDI-TOF MS (AutoFlex III, Bruker Daltonics, Germany). The sample was mixed with the MALDI matrix solution (10 mg CHCA in 1 mL 70% ACN with 0.1% TFA) on the stainless steel target and air-dried. The sample was irradiated by an Nd:YAG laser (355 nm; 3 ns pulse duration) for both desorption and ionization, with spectra recorded in the m/z range of 15,000–50,000 Da. The molecular weight of protein was processed and determined by FlexControl software (version 3.4, Bruker Daltonics, Germany, 2011).

### 5.4. Protein Concentration

Since mPEG-aldehyde has no absorbance at 280 nm and the succinimide amine reactive group does not contain UV active chromophores, UV adsorption at 280 nm is used to quantify the protein concentration [31]. A mass extinction coefficient (ε_0.1%_) of 0.278 (mg/mL protein)^−1^ cm^−1^ was obtained from ExPASy (SIB Bioinformatics Resource Portal, Lausanne, Switzerland) and used to calculate the protein concentration from the absorbance at 280 nm (NanoDrop, Thermo Fisher Scientific Inc., Waltham, MA, USA). The molar concentrations of trimucrin RR and its PEGylated variant were calculated based on their total molecular weights (RR: 8.0 kDa; PEG-RR: 40.0 kDa). 

### 5.5. Analytical Methods

Native RR and PEG-RR were analyzed by SDS-PAGE (10% polyacrylamide gradient gel). Gels were stained with Coomassie blue, and then analyzed using a gel imaging system from Invitrogen (Thermo Fisher Scientific Inc., Waltham, MA, USA). The in vitro fibrinolytic activity of both purified RR and PEG-RR were measured using the method of Asturp and Mullertz as we reported previously (27).

### 5.6. Preparation of Human PRP, PS, and Platelet Aggregation Assay

All participants provided informed consent, and this study was approved by the Institutional Review Board (17-S-032-2). Preparation of human PRP, PS, and the platelet aggregation assay were conducted as described previously [32]. The collected blood of healthy human volunteers was anticoagulated with 3.8% sodium citrate (9:1 v/v), and then immediately centrifuged at 2100 rpm for 9 min at room temperature (RT) in order to obtain PRP. For PS, blood was anticoagulated with acid citrate dextrose (ACD; 9: 1, v/v) and then centrifuged at 2100 rpm for 9 min at RT. The supernatant was supplemented with prostaglandin E_1_ (PGE_1_; 0.5 μM) and heparin (6.4 U/mL), incubated for 10 min at 37 °C and centrifuged at 3200 rpm for 8 min. After discarding the supernatant, the platelets were washed two times with Tyrode’s solution (NaCl 136.9 mM; KCI 11.9 mM; MgCl_2_ 2.1 mM; CaCl_2_ 2 mM; NaH_2_PO_4_ 0.4 mM; NaHCO_3_ 11.9 mM; glucose 11.1 mM; BSA 3.5 mg/mL; pH 7.35–7.4). Finally, the washed platelets were suspended in Tyrode’s solution and the platelet count was adjusted to 3.75 × 10^8^ (platelets/mL). Platelet aggregation was measured with a Lumi-Aggregometer (Payton Scientific, Buffalo, NY, USA) under continuous stirring at 900 rpm. Human PRP or PS was added to the silicon-coated cuvette and incubated at 37 °C with an appropriate amount of Tyrode’s buffer. Agents was added 3 min before the addition of the inducer. The degree of inhibition of platelet aggregation was expressed as a percentage of inhibition using the following equation: X(%) = (1 − B)/A × 100%, where A is the maximum aggregation of vehicle treated platelets, and B is the maximum aggregation of inhibitor treated platelets.

### 5.7. Definition of Safety Index

AP2 is an inhibitory monoclonal antibody raised against α_IIb_β_3_. In combination with AP2 (4 μg/mL), the lowest concentration of the agent to activate platelets/the IC_50_ of the agent on collagen-induced platelet aggregation was defined as the safety index [8,15].

### 5.8. Rotational Thromboelastometry (ROTEM)

ROTEM assay (Tem International, Munich, Germany) was performed according to the ROTEM^®^ delta system’s instructions as previously described [8,15]. Four variables the including maximum clot firmness (MCF, mm), clotting time (CT, sec), clot formation time (CFT, sec), and alpha angle (α, o) were incorporated into a coagulation index (CI) as defined by the equation: CI= −0.6516CT − 0.3772CFT + 0.1224MCF + 0.0759α − 7.7922. The CI function represents as an overall assessment of coagulation and primary hemostasis, with values over +3 said to be hyper-coagulable and values less than −3 said to represent a hypo-coagulable state.

### 5.9. Ethics of Animal Experiments

The animal experimental procedures were approved by the Laboratory Animal Use Committee of Mackay Medical College (A1060020; Approval Date: 07/01/2018–07/31/2021). All experiments with animals were performed according to the ARRIVE Guidelines and followed the recommendations of the NIH Guide for the Care and Use of Laboratory Animals.

### 5.10. In Vitro and ex vivo Mouse Platelet Aggregation

Male ICR was anesthetized with sodium pentobarbital (50 mg/kg, intraperitoneal). For an in vitro mouse platelet aggregation assay, blood samples were collected by an intracardiac puncture. PRP was obtained by centrifuging the blood sample at 200× *g* for 4 min and preincubated with saline, eptifibatide, RR, or PEG-RR for 3 min, and then treated with 10 μg/mL collagen (Sigma-Aldrich Chemical Co., St. Louis, MO, USA). For ex vivo mouse platelet aggregation assay, mice were treated intravenously with saline, eptifibatide, RR, or PEG-RR. Blood samples were collected at 5 min by intracardiac puncture. Mouse PRP was obtained by centrifuging the blood sample at 200× *g* for 4 min, and then treated with collagen.

### 5.11. Drug Administration

Eptifibatide, RR, or PEG-RR solution was freshly prepared before administration. For reducing the effect on the animal’s wellbeing of animals that were used in this study, the most effective dose of agents on inhibition of platelet aggregation was applied to investigate whether these agents increase bleeding risk in vivo. Agents or the vehicle (PBS) was infused via a lateral tail vein at 5 min before intracardiac puncture or tail amputation. Mice injected with a vehicle were used as controls.

### 5.12. Immune Clearance of Platelets in FcγRIIa-Transgenic Mice

FcγRIIa-transgenic mice obtained from The Jackson Laboratory [16] (weighing 24–30 g) were used in this test. FcγRIIa-transgenic mice were intravenously injected with RR or PEG-RR via a lateral caudal vein, and then blood was collected and anticoagulated with sodium citrate 5 min. The platelet amount of the whole blood samples was measured by the Sysmex cell counter (Chuo-Ku Kobe, Japan).

### 5.13. Tail-Bleeding Time

Bleeding time was measured with transection of the tail in a mouse model. A sharp cut of 2 mm from the tip of the tail was made after injection with RR or PEG-RR intravenously. The amputated tail was immediately placed in a tube filled with saline and kept at 37 °C for measuring bleeding times. The bleeding time was recorded as the arrest of bleeding for at least 10 s [33].

### 5.14. Pharmacokinetic Studies

Male ICR were intravenously administered with eptifibatide, RR, or PEG-RR at indicated concentrations. Blood was obtained via vena cava. Citrated PRP was collected by centrifugation. The FcγRIIa-transgenic mice were intravenously treated with eptifibatide, RR, or PEG-RR. At different time intervals up to 72 h after injection, whole blood (100 μL) was collected by puncture of the retro-orbital sinus of anesthetized mice using heparinized hematocrit tubes. Blood was anticoagulated with sodium citrate for 5 min, and PRP was prepared as described above. Platelet aggregation was initiated with ADP (final concentration 20 μM) and monitored in a Lumi-Aggregometer (Payton Scientific, Buffalo, NY, USA) under continuous stirring at 900 rpm [21]. The platelet inhibitory effect of agents at different time intervals was recorded in estimating the in vivo functional half-life of agents at increasing doses.

### 5.15. Irradiation-Induced Mesenteric Vessel Thrombosis Model

Fluorescent dye-induced platelet thrombus formation in mesenteric microvessels of male ICR mice (weighing 20–30 g) was performed as described previously [34,35]. After intravenous injection with saline, RR, or PEG-RR, the microvascular bed was observed using an Olympus IX71 inverted phase/fluorescence microscopes equipped with 10× objective lenses, and then recorded using an Olympus DP71 digital microscope camera (Olympus America Inc., Savage, MN, USA) and version 4.6 of the SPOT software (Micro Video Instruments Inc., Avon, MA, USA).

### 5.16. Statistical Analysis

Results were expressed as mean ± SEM. Statistical analysis was performed to determine group differences through analysis of variance. The significant difference between two groups was evaluated by a one-way analysis of variance (ANOVA) and the Newman–Keuls multiple comparison tests, and *p* < 0.05 was considered as a significant difference.

## Figures and Tables

**Figure 1 toxins-12-00426-f001:**
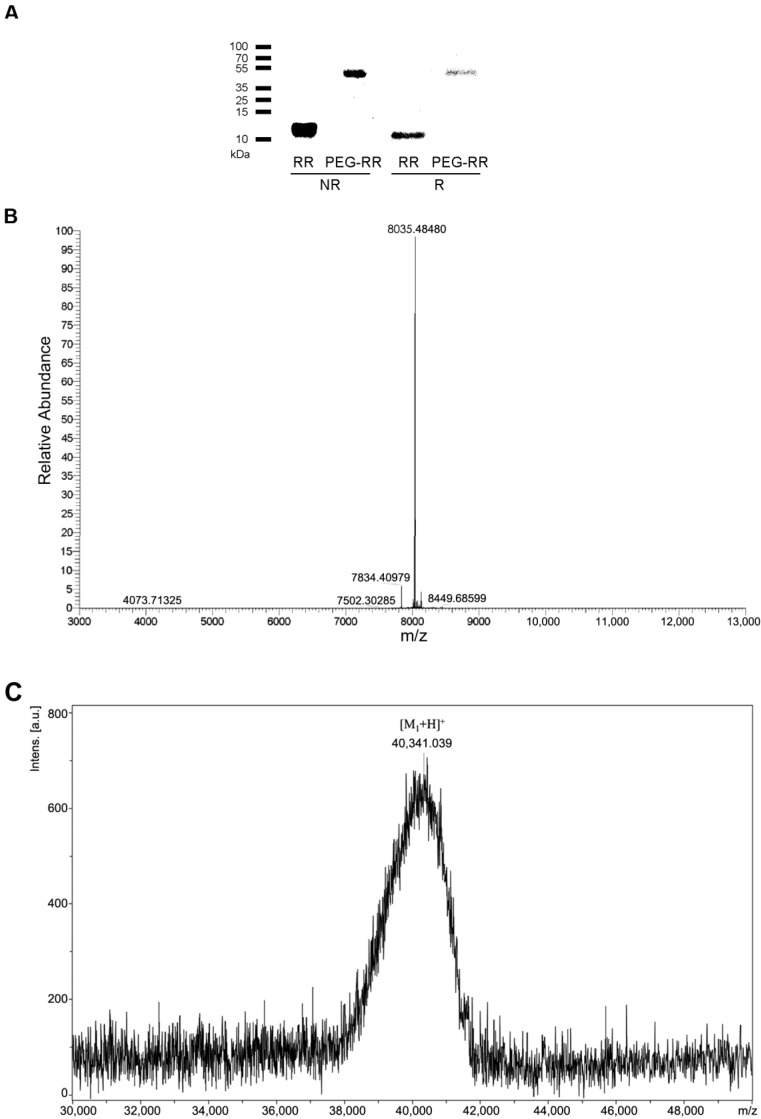
The molecular mass and purity of RR and PEG-RR. (**A**) RR and PEG-RR were subjected to SDS-PAGE under non-reduced (NR) and reduced (R) conditions. The gel was stained with Coomassie brilliant blue. (**B**,**C**) Mass spectra of RR (**B**) and PEG-RR (**C**) showed peak with molecular masses of 8035.48 Da and 40340.09 Da, respectively.

**Figure 2 toxins-12-00426-f002:**
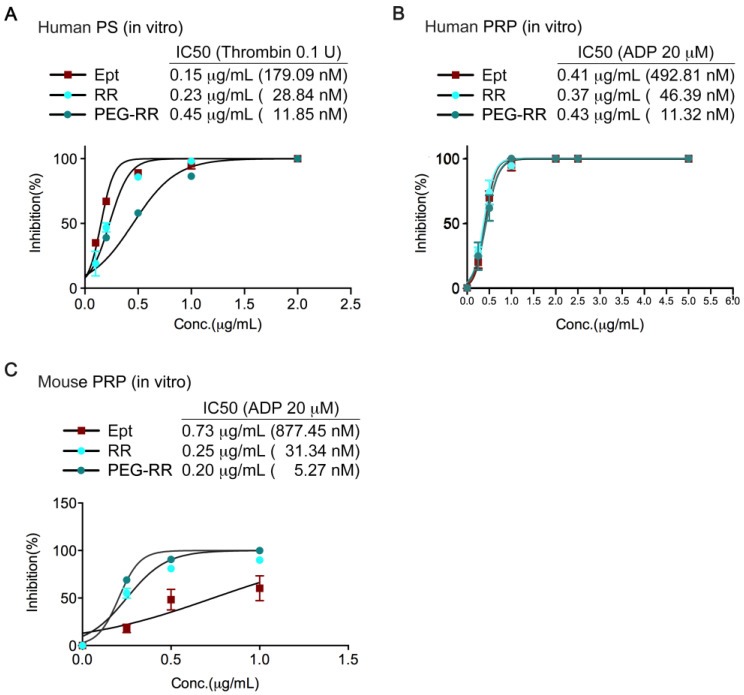
Effect of RR and PEG-RR on platelet aggregation in human platelet-rich plasma, human platelet suspension, and mouse platelet-rich plasma. (**A**) Washed human platelet suspension (PS) was preincubated with various concentrations of RR or PEG-RR at 37 °C, 3 min before the addition of thrombin (0.1 U/mL). Human (**B**) or mouse (**C**) platelet-rich plasma (PRP) was preincubated with various concentrations of RR or PEG-RR and stirred for 3 min at 37 °C before the addition of ADP (20 μM) to trigger platelet aggregation. Platelet aggregation was measured by the turbidimetric method (ΔT) using a platelet aggregometer. The data were presented as mean (*n* > 3). The molecular mass of RR and PEG-RR was estimated to be approximately 8036.07 and 37976.95 Da, respectively.

**Figure 3 toxins-12-00426-f003:**
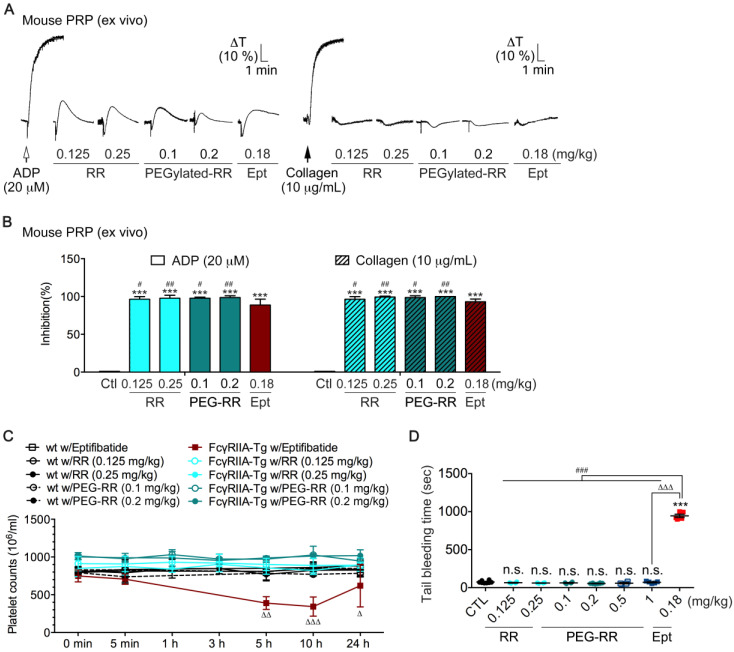
Comparison of the ex vivo and in vivo antiplatelet activity and the bleeding tendency among RR, PEG-RR, and eptifibatide. (**A**,**B**) Mice were intravenously treated with saline, RR (0.125 and 0.25 mg/kg), PEG-RR (0.1 and 0.2 mg/kg), or eptifibatide (Ept, 0.18 mg/kg), and then blood samples were collected by intracardiac puncture. Mouse platelet-rich plasma (PRP) was obtained by centrifugation at 200× *g* for 4 min, and then ADP (left, 20 μM) or collagen (right, 10 μg/mL) was added to trigger platelet aggregation. Platelet aggregation was measured by the turbidimetric method (ΔT) using a platelet aggregometer. Typical tracing curves shown (**A**) are representative of eight independent experiments. Data analytics (**B**) are presented as the percent inhibition of control. (**C**) Comparison of the immune thrombocytopenia tendency among eptifibatide, RR, and PEG-RR in an FcγRIIA transgenic mouse model. The wild-type and FcγRIIA transgenic mice were intravenously treated with eptifibatide (0.18 mg/kg), RR (0.125 and 0.25 mg/kg), or PEG-RR (0.1 and 0.2 mg/kg), and then whole blood (100 μL) was collected by puncture of the retro-orbital sinus of anesthetized mice using heparinized hematocrit tubes. Platelet counts were obtained before and at timed intervals after the injection of antithrombotic agents. Notes that immune clearance of platelets occurs in i.v. injection of eptifibatide, but not in RR and PEG-RR. (**D**) Bleeding time was measured 5 min after the intravenous injection of saline (Ctl), RR (0.125 and 0.25 mg/kg), PEG-RR (0.1, 0.2, 0.5, and 1 mg/kg), or eptifibatide (Ept, 0.18 mg/kg). The average bleeding time is indicated as (—). Each different symbol represents the bleeding time of an individual mouse. (**A**) Representative traces from at least three experiments performed in duplicate. (**B**) *n* = 8; (**C**) *n* = 6; (**D**) *n* = 5. *** *p* < 0.001; n.s., non-significance, significantly different from the control group; # *p* < 0.05, ## *p* < 0.01, and ### *p* < 0.001, significantly different from Eptifibatide group; Δ *p* < 0.05, ΔΔ *p* < 0.01, and ΔΔΔ *p* < 0.001, significantly different from PEG-RR group (paired Newman–Keuls test).

**Figure 4 toxins-12-00426-f004:**
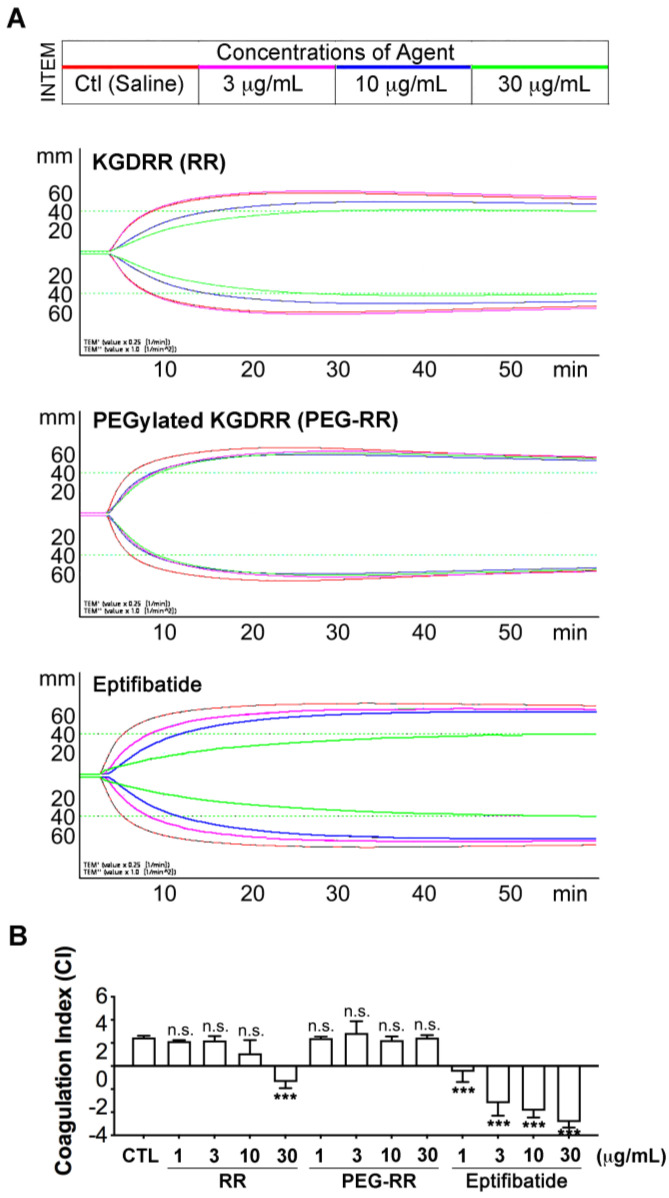
Rotational thromboelastometry assays (ROTEM) show the comparable hemostatic function of RR, PEG-RR, and eptifibatide in human whole blood. After indicated agents had been added to human blood, the clotting time, clot formation time, α-angle, and maximum clot firmness were evaluated with a ROTEM analyzer following recalcification of the blood, and determined for clot kinetics. The kinetic trace (**A**) showed the relative activity of increasing concentrations of agents on the concentration indices of human whole blood during the coagulation process. CTL (control group; red line): in the absence of agent administration. Coagulation index (**B**) was determined for clot kinetics and assessed coagulation process. *n* = 4; mean; *** *p* < 0.001, significantly different from the control group by paired Newman-Keuls test; n.s., non-significance.

**Figure 5 toxins-12-00426-f005:**
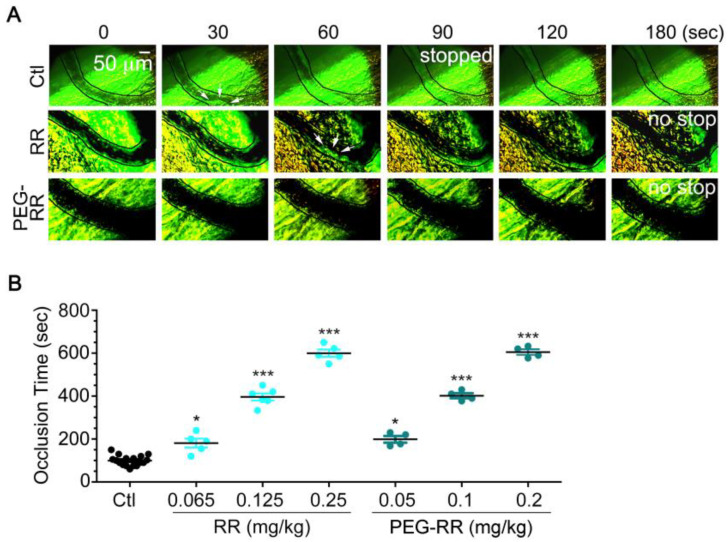
Effect of RR and PEG-RR on fluorescent dye-induced platelet-rich thrombus formation in mesenteric venules of mice. (**A**) The fluorescein sodium (12.5 mg/kg) was administered intravenously through a lateral tail vein of the mouse for 5 min. The male Institute of Cancer Research (ICR) mice were injected with saline (Ctl), RR (0.065, 0.125, and 0.25 mg/kg) or PEG-RR (0.05, 0.1, and 0.2 mg/kg) 5 min before illumination-induced venular wall injury, and then the thrombus formation of illuminated vessels was monitored by Olympus IX71 inverted phase/fluorescence. Representative images of the progression of fluorescent dye-induced platelet-rich thrombus formation in mesenteric venules of mice in the context of the bright-field microvascular histology at the indicated time points. All panels are × 100 with the same scale bar in the upper left panel; scale bar indicates 50 μm; blood flow was from left to right; Arrows indicate the thrombi which were first observed. (**B**) Time to total occlusion of the injured venules; each different symbol represents the occlusion time of the individual mouse and the average time is indicated as (—). The time to total occlusion of the injured venules are presented as mean (*n* > 3, * *p* < 0.05, *** *p* < 0.001 compared to control group by paired Newman-Keuls test).

**Figure 6 toxins-12-00426-f006:**
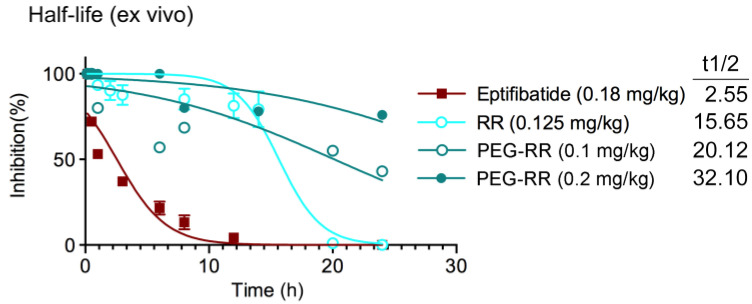
Calculated functional half-life of RR, PEG-RR, and eptifibatide in inhibiting ex vivo platelet aggregation in mouse platelet-rich plasma. Mice were intravenously treated with RR, PEG-RR (ranging 0.1–1 mg/kg), or eptifibatide and the blood samples were collected by intracardiac puncture after the indicated time intervals (0.17, 0.5, 1, 2, 6, 8, 12, and 24 h). PRP was obtained after centrifugation, and then ADP (20 μM) was added to trigger platelet aggregation. Platelet aggregation was measured by the turbidimetric method (ΔT) using a platelet aggregometer. Data are presented as inhibitory percent of aggregation in vehicle control. All the data were presented as mean (*n* > 3) and half-life of antithrombotics were calculated.

**Table 1 toxins-12-00426-t001:** IC_50_, safety index, rotational-thromboelastometry (ROTEM) variables, and tail-bleeding time of trimucrin, intact RR, PEG-RR, and the clinical anti-thrombotic agents. IC_50_ of collagen (10 μg/mL)-induced platelet aggregation in washed platelet suspension. Clotting time, clot formation time, and maximum clit firmness are evaluated via a ROTEM assay described in the method. The safety index is estimated as the lowest concentration of disintegrin to activate platelet (combining with 4 μg/mL AP2)/IC_50_ of disintegrin on collagen-induced platelet aggregation. These experiments were repeated at least five times and values were presented as means. * *p* < 0.05, ** *p* < 0.01, *** *p* < 0.001 compared with control group by Dunnett’s test; NS, non-significance. N/A, not applicable.

Antithrombotic Agents	Trimucrin		RR		PEG-RR		Tirofiban		Eptifibatide		Control
IC_50_ (nM)	58.51		36.9		8.6		85.1		625.0		N/A
Safety index	23.4		>3365.9		>3071.3		4.0		3.6		N/A
Fold of IC_50_	4×	20×	4×	20×	4×	20×	4×	20×	4×	20×	N/A
Clotting time (sec)	200.0 (NS)	224.3 (NS)	204.1 (NS)	221.5 (NS)	201.3 (NS)	196.8 (NS)	263.3 (**)	268.3 (***)	246.0 (*)	251.3 (*)	202.1
Clot formation time (sec)	96.6 (NS)	102.7 NS)	91.5 (NS)	101.9 (NS)	86.7 (NS)	93.1 (NS)	235.7 (***)	341.5 (***)	248.3 (***)	309.2 (***)	87.2
Maximum clot firmness (mm)	59.8 (NS)	55.7 (NS)	61.3 (NS)	57.4 (NS)	61.3 (NS)	60.5 (NS)	50.7 (**)	44.3 (***)	52.3 (*)	47.3 (***)	60.9
Tail-bleeding time (sec)	95.0 (NS)	190.2 (*)	63.5 (NS)	68.4 (NS)	66.2 (NS)	66.3 (NS)	480.7 (***)	836.2 (***)	538.6 (***)	799.5 (***)	66.9

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
