# Peer review of "Improved Antithrombotic Activity and Diminished Bleeding Side Effect of a PEGylated α_IIb_β_3_ Antagonist, Disintegrin"

_toxins, 2020, doi:10.3390/toxins12070426_

Round 1
Reviewer 1 Report
Manuscript: ID 798360:
Improved antithrombotic activity and diminished bleeding side effect of a PEGylated αIIbβ3 antagonist, disintegrin
This is an interesting study for antithrombotic activity using pegylated αIIbβ3 antagonists. However few major comments need to be addressed before accepting for publication.
- The authors mutated trimucrin to generate the RR sequence used in the study. It was mentioned in the introduction that trimucrin has a better safety profile and longer circulating half-life than eptifibatide. Hence the rationale of selecting eptifibatide to compare the efficacy of the RR and PEG- RR was not justifiable. The authors need to clarify if there is any justification.
- The basics of the study area is αIIbβ3 antagonist. Since the authors mutated the sequence, some data should be provided to show the effect is mediated through αIIbβ3 antagonism or by any other mechanism.
- Fig 1: The authors should include mass spectra for PEG-RR.
- Figure 2A-C. It would be better if the x-axis values in graphs A, B, and C kept constant.
- How the authors calculated the concentration of PEG-RR? Details should be added. Is it based on total molecular weight or based on the RR present in PEG-RR. This information is important.
- There is no description of Figure 3A such as what is the analysis, and how to read the data, etc. If this is not relevant for the paper better to remove it.
- Figure 3B ex vivo and collagen and fig 3C and or D (Tail bleeding time): The concentrations used for RR are 0.125 and 0.25mg/Kg, PEG-RR is 0.1 and 0.2mg/KG and Ept is 0.18mg/KG. There is no real comparison can be done between the groups based on the concentrations. The authors need to explain the rationale for selecting such concentration differences between the groups.
- The statistical significance of the results in Figure 3B-D was calculated compared to control. The significance between the groups RR and PEG RR as well as compared to Ept should provide.
- Also please check the figure number Figure 3C and D. In the text tail bleeding time is reported as Figure 3C. Need to correct it.
- The page 7 Line 169mentioned 1mg/Kg of PEG-RR. Any data pertaining to that concentration is not in the manuscript. Please check it. It is not clear what the authors trying to express in that sentence.
- There is no data provided for Ept in ROTEM data in figure 4 and section 2.5 (Figure 5A-B). Update the figure with Ept data.
- Please include the details of all chemicals used such as catalog number, molecular weight, etc. For example, there was no mention of what molecular weight PEG was used in this study.
- Methodology: The authors need to focus more on the methodology section. It looks like in all sections the authors mentioned a reference to say that you followed that methodology. It will not help the reviewer or the readers. A clear description of the methodology (even though followed established protocol) should be included particularly in section 5.4, 5.5, 5.6 5.9, etc.
- References: There is only one reference from 2016, two from 2017, and one from 2019. This is not a clear picture of the αIIbβ3 antagonist study from the last 5 years. The authors should update the reference by including more recent literature in the discussion.
Reviewer 2 Report
MAJOR COMMENTS
Comment 1. It is not clear in the abstract that KGDRR is a timucrin derivative and not only the small sequence KGDRR. I would suggest to use another nomenclature, such as "[KGDRR]timucrin", to make clear that this is not a small peptide but a timucrin variant.
Comment 2. Table 1, statistically a mean of three repeats is not valid. Please show the range or each individual values. In the "Clot formation", the table mentions that the difference between control and the 20x of trimucrin, KGDRR and PEG-KGDRR are "non-significant", but the difference is higher or equal to the values of Tirofiblan x4 that is considered to be significant. Please reinterpret these data appropriately.
Comment 3. According to data Table 1, the compounds KGDRR and PEG-KGDRR have lower potency than Eptifibatide and Tirofiban (as equal concentration), please commment and mentions this in the abstract.
Comment 4. What is the in vivo half life of the pegylated and non-pegylated [KGDRR]timucrin? The authors evaluated the functional half life but not the actual stability of the compounds. I am concern that the RR would be easily cleaved in vivo. The authors mention in ines 260 that the peptides had "prolonged biological half-life" but they did not prove it.
MINOR COMMENTS
Comment 5. On line 80, the authors talk about safety index. Please give details on the definition of the index.
Comment 6. Figure 4 (B). Please specify what the "***" means.
Round 2
Reviewer 1 Report
NA
Author Response
Thank you for your comment.
The manuscript with our changes in red has been revised to improve its English language and grammar.
Hope the revised version may meet the requirement that you suggested.